# Tighter sparse variational Gaussian processes

**Thang D. Bui**  *thang.bui@anu.edu.au*
*School of Computing*
*Australian National University*

**Matthew Ashman**  *mca39@cam.ac.uk*
*Department of Engineering*
*University of Cambridge*

**Richard E. Turner**  *ret26@cam.ac.uk*
*Department of Engineering*
*University of Cambridge*

**Reviewed on OpenReview:** *https://openreview.net/forum?id=L33DSu3zvq*

## Abstract

Sparse variational Gaussian process (GP) approximations based on inducing points have become the de facto standard for scaling GPs to large datasets, owing to their theoretical elegance, computational efficiency, and ease of implementation. This paper introduces a provably tighter variational approximation by relaxing the standard assumption that the conditional approximate posterior given the inducing points must match that in the prior. The key innovation is to modify the conditional posterior to have smaller variances than that of the prior at the training points. We derive the collapsed bound for the regression case, describe how to use the proposed approximation in large data settings, and discuss its application to handle orthogonally structured inducing points and GP latent variable models. Extensive experiments on regression benchmarks, classification, and latent variable models demonstrate that the proposed approximation consistently matches or outperforms standard sparse variational GPs while maintaining the same computational cost.

## 1 Introduction

Gaussian processes (GPs) (Rasmussen & Williams, 2006) provide a powerful framework for modelling probability distributions over functions, offering principled uncertainty quantification and ease of use. Their flexibility in encoding domain knowledge—such as smoothness, peridocity, or domain-specific structure—has led to widespread adoption across scientific and engineering applications. Exact inference in GP models poses significant computational challenges, requiring $\mathcal{O}(N^3)$ time and $\mathcal{O}(N^2)$ space complexity for $N$ observations. A suite of approximations have been developed to address these limitations. Most notably, sparse variational Gaussian processes (SVGP; Titsias, 2009; Hensman et al., 2015; Matthews et al., 2016) address the poor computational complexity through the use of an approximate posterior distribution parameterised by a small set of *inducing points*.

The standard SVGP framework employs a structured variational approximation that factorises the posterior distribution over the unknown function $f$ into two components: $q(f) = p(f_{\neq u}|u)q(u)$. Here, $p(f_{\neq u}|u)$ represents the GP prior distribution conditioned on the function values at inducing locations $z$, $u = f(z)$. The second term, $q(u)$, is modelled as a multivariate Gaussian distribution. Improved variational approximations have been developed—such as SOLVE-GP (Shi et al., 2020)—which use more sophisticated distributions for $q(u)$.

This paper introduces a novel approach to improving SVGP approximations by modifying the conditional GP prior distribution at observed inputs, rather than focusing solely on the inducing point distribution. For

Gaussian likelihoods, our approach yields a new and improved collapsed lower bound on the log marginal likelihood that involves *no additional variational parameters*. Furthermore, we show how the uncollapsed form of our bound facilitates the use of stochastic mini-batch optimisation and extends naturally to non-Gaussian likelihoods through a single additional variational parameter. We demonstrate the versatility of our method by integrating it with SOLVE-GP and extending it to sparse variational approximations in the GP latent variable model (GPLVM; Lawrence, 2005; Damianou et al., 2016). Our results demonstrate that by targeting our improved lower bound, our approach consistently improves the predictive performance and log marginal likelihood estimates across a range of regression, classification, and latent variable modelling tasks. An implementation is made available in this repository `https://github.com/thangbui/tighter_sparse_gp`. We note that the concurrent work of Titsias (2025) presents an identical approximation but offers a different exposition to this paper.

## 2 Background

This section provides a concise introduction to pseudo-point based sparse variational Gaussian processes (SVGP; Titsias, 2009; Hensman et al., 2015; Matthews et al., 2016; Bui et al., 2017). Consider GP regression with Gaussian observation noise:

$$p(f|\gamma) = \mathcal{GP}(f; 0, k_\gamma), \tag{1}$$

$$p(\boldsymbol{y}|f, \boldsymbol{x}, \sigma^2) = \mathcal{N}(\boldsymbol{y}; f(\boldsymbol{x}), \sigma^2 \mathbf{I}), \tag{2}$$

where $\boldsymbol{x} \in \mathbb{R}^{N \times D}$ and $\boldsymbol{y} \in \mathbb{R}^N$ are the training inputs and corresponding noisy outputs, $f$ denotes the unknown function mapping from input to output, $k_\gamma$ is the covariance function governed by hyperparameters $\gamma$, and $\sigma^2$ is the observation noise. These hyperparameters, denoted collectively as $\theta$, can be found by maximising the log marginal likelihood:

$$\mathcal{L}(\theta) = -\frac{N}{2}\log(2\pi) - \frac{1}{2}\boldsymbol{y}^\intercal (\mathbf{K_{ff}} + \sigma^2 \mathbf{I})^{-1} \boldsymbol{y} - \frac{1}{2}\log|\mathbf{K_{ff}} + \sigma^2 \mathbf{I}|, \tag{3}$$

where $\mathbf{K_{ff}}$ is the covariance between training function values $\boldsymbol{f} = f(\boldsymbol{x})$. This objective takes $\mathcal{O}(N^3)$ to compute and is thus computationally prohibitive for large $N$. To sidestep this, we use an approximate posterior judiciously parameterised by a small set of pseudo-points or inducing points as follows:

$$q(f) = p(f_{\neq \boldsymbol{f}, \boldsymbol{u}}|\boldsymbol{f}, \boldsymbol{u})p(\boldsymbol{f}|\boldsymbol{u})q(\boldsymbol{u}), \tag{4}$$

where $\boldsymbol{u} = f(\boldsymbol{z}) \in \mathbb{R}^M$ and $\boldsymbol{z} \in \mathbb{R}^{M \times D}$ are the inducing outputs and inputs, respectively, and $M \ll N$. The conditional $q(f_{\neq \boldsymbol{u}}|\boldsymbol{u})$ in the approximate posterior is chosen to match that in the prior, leading to the following variational objective,

$$\mathcal{F}_0(q(\boldsymbol{u}), \theta) = \left\langle \log \frac{p(f)p(\boldsymbol{y}|f, \boldsymbol{x})}{q(f)} \right\rangle_{q(f)} = \left\langle \frac{\log \cancel{p(f_{\neq \boldsymbol{f}, \boldsymbol{u}}|\boldsymbol{f}, \boldsymbol{u})}\cancel{p(\boldsymbol{f}|\boldsymbol{u})}p(\boldsymbol{u})p(\boldsymbol{y}|f, \boldsymbol{x})}{\cancel{p(f_{\neq \boldsymbol{f}, \boldsymbol{u}}|\boldsymbol{f}, \boldsymbol{u})}\cancel{p(\boldsymbol{f}|\boldsymbol{u})}q(\boldsymbol{u})} \right\rangle_{q(f)}$$

$$= -\mathrm{KL}[q(\boldsymbol{u})||p(\boldsymbol{u})] + \sum_n \int_{\boldsymbol{u}, f(x_n)} q(\boldsymbol{u})p(f(x_n)|\boldsymbol{u})\log p(y_n|f(x_n)). \tag{5}$$

Titsias (2009) showed that when the likelihood is Gaussian, an analytic optimal form for $q(\boldsymbol{u})$ can be found, $q(\boldsymbol{u}) \propto p(\boldsymbol{u})\mathcal{N}(\boldsymbol{y}; \mathbf{K_{fu}}\mathbf{K_{uu}^{-1}}\boldsymbol{u}, \sigma^2 \mathbf{I})$, and that a collapsed bound is also analytically available,

$$\mathcal{F}_1(\theta) = -\frac{N}{2}\log(2\pi) - \frac{1}{2}\boldsymbol{y}^\intercal(\mathbf{Q_{ff}} + \sigma^2 \mathbf{I})^{-1}\boldsymbol{y} - \frac{1}{2}\log|\mathbf{Q_{ff}} + \sigma^2 \mathbf{I}| - \frac{1}{2\sigma^2}\mathrm{trace}(\mathbf{D_{ff}}), \tag{6}$$

where $\mathbf{Q_{ff}} = \mathbf{K_{fu}}\mathbf{K_{uu}^{-1}}\mathbf{K_{uf}}$ and $\mathbf{D_{ff}} = \mathbf{K_{ff}} - \mathbf{Q_{ff}}$. Crucially, the bound above can be computed in $\mathcal{O}(NM^2)$. The non-collapsed bound in eq. (5) is amenable to non-Gaussian likelihoods and data mini-batch settings (see e.g., Hensman et al., 2015), further reducing the training computational complexity to $\mathcal{O}(BM^2 + M^3)$ where $B$ is the mini-batch size. Due to this small complexity and the ease of implementation, the above variational approach has arguably become the go-to sparse approximation in the GP literature. In this work, we will revisit its core assumption of matching prior and posterior conditionals and show that relaxing this assumption results in a tighter and more performant approximation.

## 3   A tighter variational approximation

The variational approximation in eq. (4) is chosen such that the conditional $q(\boldsymbol{f}|\boldsymbol{u})$ identically matches the prior conditional $p(\boldsymbol{f}|\boldsymbol{u})$. Instead, we propose using the following variational posterior,

$$q(f) = p(f_{\neq \boldsymbol{f},\boldsymbol{u}}|\boldsymbol{f},\boldsymbol{u})q(\boldsymbol{f}|\boldsymbol{u})q(\boldsymbol{u}), \tag{7}$$

where $q(\boldsymbol{f}|\boldsymbol{u}) = \mathcal{N}(\boldsymbol{f}; \mathbf{K_{fu}}\mathbf{K_{uu}^{-1}}\boldsymbol{u}; \mathbf{D_{ff}^{1/2}}\mathbf{M}\mathbf{D_{ff}^{\top/2}})$, $\mathbf{M}$ is a diagonal matrix, $\mathbf{M} = \text{diag}([m_1, m_2, \ldots, m_N])$ and $m_n > 0$. Note that the mean of the prior conditional $p(\boldsymbol{f}|\boldsymbol{u}) = \mathcal{N}(\boldsymbol{f}; \mathbf{K_{fu}}\mathbf{K_{uu}^{-1}}\boldsymbol{u}; \mathbf{D_{ff}})$ is retained in $q(\boldsymbol{f}|\boldsymbol{u})$. The resulting variational bound is,

$$\begin{aligned}
\mathcal{F}_2(q(\boldsymbol{u}), \theta, \mathbf{M}) &= \left\langle \log \frac{\cancel{p(f_{\neq \boldsymbol{f},\boldsymbol{u}}|\boldsymbol{f},\boldsymbol{u})}p(\boldsymbol{f}|\boldsymbol{u})p(\boldsymbol{u})p(\boldsymbol{y}|f,\boldsymbol{x})}{\cancel{p(f_{\neq \boldsymbol{f},\boldsymbol{u}}|\boldsymbol{f},\boldsymbol{u})}q(\boldsymbol{f}|\boldsymbol{u})q(\boldsymbol{u})} \right\rangle_{q(f)} \\
&= -\text{KL}[q(\boldsymbol{u})||p(\boldsymbol{u})] - \int_{\boldsymbol{u}} q(\boldsymbol{u})\text{KL}[q(\boldsymbol{f}|\boldsymbol{u})||p(\boldsymbol{f}|\boldsymbol{u})] \\
&\quad + \sum_n \int_{\boldsymbol{u},f(x_n)} q(\boldsymbol{u})q(f(x_n)|\boldsymbol{u}) \log p(y_n|f(x_n)).
\end{aligned} \tag{8}$$

Due to the structure of the variational distribution, the middle term can be simplified to,

$$-\int_{\boldsymbol{u}} q(\boldsymbol{u})\text{KL}[q(\boldsymbol{f}|\boldsymbol{u})||p(\boldsymbol{f}|\boldsymbol{u})] = -\frac{1}{2}\text{trace}(\mathbf{M}) + \frac{1}{2}\log|\mathbf{M}| + \frac{N}{2} = \frac{1}{2}\sum_n [1 + \log(m_n) - m_n]$$

**Collapsed bound and optimal M**   In the regression case, similar to the Titsias' bound, we can obtain the optimal form for $q(\boldsymbol{u}) \propto p(\boldsymbol{u})\mathcal{N}(\boldsymbol{y}; \mathbf{K_{fu}}\mathbf{K_{uu}^{-1}}\boldsymbol{u}, \sigma^2\mathbf{I})$, leading to the following collapsed bound,

$$\mathcal{F}_3(\theta, \mathbf{M}) = -\frac{N}{2}\log(2\pi) - \frac{1}{2}\boldsymbol{y}^{\mathsf{T}}(\mathbf{Q_{ff}} + \sigma^2\mathbf{I})^{-1}\boldsymbol{y} - \frac{1}{2}\log|\mathbf{Q_{ff}} + \sigma^2\mathbf{I}| - \frac{1}{2}\sum_n \left[\frac{m_n d_n}{\sigma^2} - 1 - \log(m_n) + m_n\right]$$

Setting the partial derivatives of $\mathcal{F}_3(\theta, \mathbf{M})$ wrt $m_n$ to 0, we arrive at $m_n = \frac{\sigma^2}{d_n + \sigma^2}$ and the following bound,

$$\mathcal{F}_4(\theta) = -\frac{N}{2}\log(2\pi) - \frac{1}{2}\boldsymbol{y}^{\mathsf{T}}(\mathbf{Q_{ff}} + \sigma^2\mathbf{I})^{-1}\boldsymbol{y} - \frac{1}{2}\log|\mathbf{Q_{ff}} + \sigma^2\mathbf{I}| - \frac{1}{2}\sum_n \log\left(1 + \frac{d_n}{\sigma^2}\right), \tag{9}$$

where $d_n$ is the $n$-th element in the diagonal of $\mathbf{D_{ff}}$, $d_n = k_{f_n f_n} - \mathbf{k}_{f_n\mathbf{u}}\mathbf{K_{uu}^{-1}}\mathbf{k}_{\mathbf{u}f_n}$.

**Comparison to Titsias' bound**   When $\mathbf{M}$ is the identity matrix, that is $m_n = 1\ \forall n$, the approximation in eq. (7) becomes the Titsias' variational approximation in eq. (4) and the bound in $\mathcal{F}_3(\theta)$ becomes the Titsias' bound $\mathcal{F}_1(\theta)$ in eq. (6). We note that that $\mathcal{F}_4(\theta)$ is tighter than $\mathcal{F}_1(\theta)$ due to the inequality $\log(1+x) < x$ for all $x > -1$. Our solution improves upon the solution to the Titsias' bound by allowing the marginals of the conditional approximate posterior, $q(f(x_n)|\boldsymbol{u})$, to have smaller variances than that of the conditional prior, since the optimal $m_n = \frac{\sigma^2}{d_n + \sigma^2} < 1$. Intuitively, this reduces the strength of the coupling between $q(\boldsymbol{u})$ and $q(\boldsymbol{f})$, enabling $q(\boldsymbol{f})$ to more freely model the data whilst allowing $q(f)$ to be close to the prior elsewhere.

It is also worth noting that the middle term of our bound is always non-positive. One might think that adding this term to the bound would give a poorer approximation, yet, the improvement in the expected log-likelihood (due to the smaller predictive variances at the training points—see predictions below) can yield a larger improvement to counteract.

**Stochastic mini-batch settings**   The new bound can also handle data mini-batching, yielding an *unbiased* estimator of the uncollapsed bound in eq. (8) as follows,

$$\begin{aligned}
\mathcal{F}_2(q(\boldsymbol{u}), \theta, \mathbf{M}) &\approx -\text{KL}[q(\boldsymbol{u})||p(\boldsymbol{u})] + \frac{N}{2B}\sum_b [1 + \log(m_b) - m_b] \\
&\quad + \frac{N}{B}\sum_B \int_{\boldsymbol{u},f(x_b)} q(\boldsymbol{u})q(f(x_b)|\boldsymbol{u}) \log p(y_b|f(x_b)).
\end{aligned} \tag{10}$$

**Non-Gaussian likelihoods and $m_n$ parameterisation**  One can parameterise $m_n$'s to satisfy their positive constraint and optimise them directly at the cost of having $N$ extra parameters. However, the optimal form for $m_n$ in the Gaussian likelihood setting suggests a more efficient parameterisation $m_n = \beta/(d_n + \beta)$ with $\beta > 0$ shared across all data points. We will use the latter parameterisation for all of our experiments.

**Predictions**  The predictive mean and variance of the predictive distribution at a test input $x_*$ are

$$m_* = \mathbf{k}_{*\mathbf{u}}\mathbf{K}_{\mathbf{uu}}^{-1}\mathbf{m_u}, \tag{11}$$

$$v_* = \mathbf{k}_{**} - \mathbf{k}_{*\mathbf{u}}\mathbf{K}_{\mathbf{uu}}^{-1}\mathbf{k}_{\mathbf{u}*} + \mathbf{k}_{*\mathbf{u}}\mathbf{K}_{\mathbf{uu}}^{-1}\mathbf{S_u}\mathbf{K}_{\mathbf{uu}}^{-1}\mathbf{k}_{\mathbf{u}*} - (\mathbf{k}_{*\mathbf{f}} - \mathbf{Q}_{*\mathbf{f}})\mathbf{V}_{\mathbf{ff}}(\mathbf{k}_{\mathbf{f}*} - \mathbf{Q}_{\mathbf{f}*}), \tag{12}$$

where $\mathbf{V}_{\mathbf{ff}} = \mathbf{D}_{\mathbf{ff}}^{-\top/2}(\mathbf{I} - \mathbf{M})\mathbf{D}_{\mathbf{ff}}^{-1/2}$. Note that (i) we can compute the predictive mean at the same cost as previous sparse approximations, and (ii) the predictive variance at a training point can be approximated by $v_n = m_n d_n + \mathbf{k}_{f_n\mathbf{u}}\mathbf{K}_{\mathbf{uu}}^{-1}\mathbf{S_u}\mathbf{K}_{\mathbf{uu}}^{-1}\mathbf{k}_{\mathbf{u}f_n}$. More generally, the variance at a new input that is not a training or inducing input is expensive due to the presence of $\mathbf{D}_{\mathbf{ff}}$ in the last term. One path to address this could be to approximate $\mathbf{D}_{\mathbf{ff}}$ by its diagonal matrix or to use only a subset of training points for this computation. However, we find that simply ignoring the last term at test time does not impact the predictive performance while substantially reducing the prediction cost (see section 6.2).

**Connections to existing bounds**  We can use the log-sum inequality[1] to bound the last term of our collapsed bound:

$$\sum_{n=1}^{N} \log\left(1 + \frac{d_n}{\sigma^2}\right) \leq N \log \frac{\left[\sum_{n=1}^{N}\left(1 + \frac{d_n}{\sigma^2}\right)\right]}{N} = N \log\left[1 + \frac{\text{trace}(\mathbf{K}_{\mathbf{ff}} - \mathbf{Q}_{\mathbf{ff}})}{N\sigma^2}\right]. \tag{13}$$

Thus a looser collapsed bound can be obtained:

$$\mathcal{F}_5(\theta) = -\frac{N}{2}\log(2\pi) - \frac{1}{2}\boldsymbol{y}^{\top}(\mathbf{Q}_{\mathbf{ff}} + \sigma^2\mathbf{I})^{-1}\boldsymbol{y} - \frac{1}{2}\log|\mathbf{Q}_{\mathbf{ff}} + \sigma^2\mathbf{I}| - \frac{N}{2}\log\left(1 + \frac{\text{trace}(\mathbf{K}_{\mathbf{ff}} - \mathbf{Q}_{\mathbf{ff}})}{N\sigma^2}\right).$$

This bound was derived by Artemev et al. (2021) based on bounds of the quadratic and log-determinant terms in the exact log marginal likelihood. This is also tighter than the Titsias' bound, that is $\mathcal{F}_4(\theta) \geq \mathcal{F}_5(\theta) \geq \mathcal{F}_1(\theta)$.

One can also view the proposed variational approximation as an instance of the sparse orthogonal approach of Shi et al. (2020) in which there are two sets of inducing points $\boldsymbol{u}$ and $\boldsymbol{v}$ with $\boldsymbol{v} := \boldsymbol{f}$, $\mathbf{m}_{\boldsymbol{v}} := \mathbf{0}$ and $\mathbf{S}_{\boldsymbol{v}} := \mathbf{D}_{\mathbf{ff}}^{1/2}\mathbf{M}\mathbf{D}_{\mathbf{ff}}^{\top/2}$. However, this view does not suggest new insights or potential improvements. We will next discuss how to use the proposed variational approximation to improve the sparse orthogonal approach and in the latent variable settings.

## 4  Application to sparse orthogonal variational GPs

The sparse orthogonal approach (SOLVEGP) of Shi et al. (2020) can be viewed as a structured approximation with two sets of pseudo-points $\boldsymbol{u}$ and $\boldsymbol{v}$,

$$q(f) = p(f_{\neq\boldsymbol{f},\boldsymbol{u},\boldsymbol{v}}|\boldsymbol{f},\boldsymbol{u},\boldsymbol{v})p(\boldsymbol{f}|\boldsymbol{u},\boldsymbol{v})q(\boldsymbol{u},\boldsymbol{v}),$$

$$q(\boldsymbol{u},\boldsymbol{v}) = \mathcal{N}(\boldsymbol{u};\mathbf{m_u},\mathbf{S_u})\mathcal{N}(\boldsymbol{v};\mathbf{K}_{\mathbf{vu}}\mathbf{K}_{\mathbf{uu}}^{-1}\boldsymbol{u} + \mathbf{m}_{\boldsymbol{v}},\mathbf{S}_{\boldsymbol{v}})$$

$$= \mathcal{N}\left(\begin{bmatrix}\boldsymbol{u}\\\boldsymbol{v}\end{bmatrix};\begin{bmatrix}\mathbf{m_u}\\\mathbf{K}_{\mathbf{vu}}\mathbf{K}_{\mathbf{uu}}^{-1}\mathbf{m_u} + \mathbf{m}_{\boldsymbol{v}}\end{bmatrix},\begin{bmatrix}\mathbf{S_u} & \mathbf{S_u}\mathbf{K}_{\mathbf{uu}}^{-1}\mathbf{K}_{\mathbf{uv}}\\\mathbf{K}_{\mathbf{vu}}\mathbf{K}_{\mathbf{uu}}^{-1}\mathbf{S_u} & \mathbf{S}_{\boldsymbol{v}} + \mathbf{K}_{\mathbf{vu}}\mathbf{K}_{\mathbf{uu}}^{-1}\mathbf{S_u}\mathbf{K}_{\mathbf{uu}}^{-1}\mathbf{K}_{\mathbf{uv}}\end{bmatrix}\right),$$

---

[1]For non-negative numbers $a_1, a_2, \ldots, a_n$ and $b_1, b_2, \ldots, b_n$, $\sum_{i=1}^{n} a_i \log \frac{a_i}{b_i} \geq \left(\sum_{i=1}^{n} a_i\right) \log \frac{\sum_{i=1}^{n} a_i}{\sum_{i=1}^{n} b_i}$ with equality iff $a_i/b_i = \text{constant}$.

where $(\mathbf{m_u}, \mathbf{S_u})$ and $(\mathbf{m_v}, \mathbf{S_v})$ are the mean and covariance variational parameters. This approximation brings computational benefits over naively using a single set of pseudo-points with cardinality $M = M_{\boldsymbol{u}} + M_{\boldsymbol{v}}$ while matching the latter's performance. We will show that the same trick used for sparse variational GPs—relaxing the conditional matching assumption $q(\boldsymbol{f}|\boldsymbol{u},\boldsymbol{v}) = p(\boldsymbol{f}|\boldsymbol{u},\boldsymbol{v})$— can improve SOLVEGP. In particular, similar to SVGP, we will use $q(\boldsymbol{f}|\boldsymbol{u},\boldsymbol{v}) = \mathcal{N}(\boldsymbol{f}; \mathbf{K_{f,uv}}\mathbf{K_{uv,uv}^{-1}}[\boldsymbol{u}^\intercal, \boldsymbol{v}^\intercal]^\intercal; \mathbf{D_{ff}^{1/2}}\mathbf{M}\mathbf{D_{ff}^{\top/2}})$ where $\mathbf{D_{ff}} = \mathbf{K_{ff}} - \mathbf{K_{f,uv}}\mathbf{K_{uv,uv}^{-1}}\mathbf{K_{uv,f}}$, $\mathbf{M}$ is a diagonal matrix, $\mathbf{M} = \mathrm{diag}([m_1, m_2, \ldots, m_N])$ and $m_n > 0$. The resulting variational bound is

$$\mathcal{F}_6(q(\boldsymbol{u},\boldsymbol{v}),\theta,\mathbf{M}) = \left\langle \log \frac{\overline{p(f_{\neq \boldsymbol{f},\boldsymbol{u},\boldsymbol{v}}|\boldsymbol{f},\boldsymbol{u},\boldsymbol{v})}p(\boldsymbol{f}|\boldsymbol{u},\boldsymbol{v})p(\boldsymbol{u},\boldsymbol{v})p(\boldsymbol{y}|f,\boldsymbol{x})}{\underline{p(f_{\neq \boldsymbol{f},\boldsymbol{u},\boldsymbol{v}}|\boldsymbol{f},\boldsymbol{u},\boldsymbol{v})}q(\boldsymbol{f}|\boldsymbol{u},\boldsymbol{v})q(\boldsymbol{u},\boldsymbol{v})} \right\rangle_{q(f)}$$

$$= -\mathrm{KL}[q(\boldsymbol{u})\|p(\boldsymbol{u})] - \mathrm{KL}[\tilde{q}(\boldsymbol{v})\|\tilde{p}(\boldsymbol{v})] + \frac{1}{2}\sum_n [1 + \log(m_n) - m_n]$$

$$+ \sum_n \int_{\boldsymbol{u},\boldsymbol{v},f(x_n)} q(\boldsymbol{u},\boldsymbol{v})q(f(x_n)|\boldsymbol{u},\boldsymbol{v})\log p(y_n|f(x_n))., \tag{14}$$

where $\tilde{q}(\boldsymbol{v}) = \mathcal{N}(\boldsymbol{v}; \mathbf{m_v}, \mathbf{S_v})$, $\tilde{p}(\boldsymbol{v}) = \mathcal{N}(\boldsymbol{v}; \mathbf{0}, \mathbf{C_{vv}})$, and $\mathbf{C_{vv}} = \mathbf{K_{vv}} - \mathbf{K_{vu}}\mathbf{K_{uu}^{-1}}\mathbf{K_{uv}}$. Note that the predictive distribution at a training point can be approximated efficiently, $q(f(x_n)) \approx \mathcal{N}(f(x_n); m_n, v_n)$ with

$$m_n = \mathbf{k}_{f_n\boldsymbol{u}}\mathbf{K_{uu}^{-1}}\mathbf{m_u} + \mathbf{c}_{f_n\boldsymbol{v}}\mathbf{C_{vv}^{-1}}\mathbf{m_u},$$

$$v_n = m_n(\mathbf{c}_{f_nf_n} - \mathbf{c}_{f_n\boldsymbol{v}}\mathbf{C_{vv}^{-1}}\mathbf{c}_{\boldsymbol{v}f_n}) + \mathbf{k}_{*\boldsymbol{u}}\mathbf{K_{uu}^{-1}}\mathbf{S_u}\mathbf{K_{uu}^{-1}}\mathbf{k}_{\boldsymbol{u}*} + \mathbf{c}_{f_n\boldsymbol{v}}\mathbf{C_{vv}^{-1}}\mathbf{S_v}\mathbf{C_{vv}^{-1}}\mathbf{c}_{f_n\boldsymbol{v}}$$

where $\mathbf{c}_{ab} = \mathbf{k}_{ab} - \mathbf{k}_{a\boldsymbol{u}}\mathbf{K_{uu}^{-1}}\mathbf{k}_{\boldsymbol{u}b}$. Similar to SVGP, the predictive variance at a new test point is expensive due to the dependence on all training points. However, similar to the tighter approximation in section 3, we found that simply ignoring this difficult term works well in practice.

## 5 Application to Bayesian GP latent variable models

Consider a GP latent variable model (GPLVM; Lawrence, 2005) with Gaussian observation noise:

$$p(\boldsymbol{x}) = \mathcal{N}(\boldsymbol{x}; 0, \mathbf{I}),$$
$$p(f|\gamma) = \mathcal{GP}(f; 0, k_\gamma),$$
$$p(\boldsymbol{y}|f, \boldsymbol{x}) = \mathcal{N}(\boldsymbol{y}; f(\boldsymbol{x}), \sigma^2\mathbf{I}).$$

Both the posterior $p(f|\boldsymbol{y})$ and marginal likelihood $p(\boldsymbol{y})$ are intractable. Instead, we introduce an approximate posterior of the following form:

$$q(f, \boldsymbol{x}) = q(\boldsymbol{x})p(f_{\neq \boldsymbol{f},\boldsymbol{u}}|\boldsymbol{f},\boldsymbol{u})q(\boldsymbol{f}|\boldsymbol{u},\boldsymbol{x})q(\boldsymbol{u})$$

$$q(\boldsymbol{f}|\boldsymbol{u},\boldsymbol{x}) = \mathcal{N}(\boldsymbol{f}; \mathbf{K_{fu}}\mathbf{K_{uu}^{-1}}\boldsymbol{u}, \mathbf{D_{ff}^{1/2}}\mathbf{M}(\boldsymbol{x})\mathbf{D_{ff}^{\top/2}}),$$

where $\mathbf{M}(\boldsymbol{x}) = \mathrm{diag}([m_1(x_1), m_2(x_2), \ldots, m_N(x_N)])$ and $m_n(x_n) > 0$. Note that $q(\boldsymbol{f}|\boldsymbol{u},\boldsymbol{x})$ depends on $\boldsymbol{x}$ through $\mathbf{K_{fu}}$, $\mathbf{D_{ff}}$ and $\mathbf{M}(\boldsymbol{x})$, and that when $\mathbf{M}$ is the identity matrix, that is $m_n = 1$, we obtain the variational approximation of Damianou et al. (2016). We can bound the log marginal likelihood as

$$\mathcal{F}(q(f, \boldsymbol{x}), \theta) = -\mathrm{KL}[q(\boldsymbol{x})\|p(\boldsymbol{x})] - \mathrm{KL}[q(\boldsymbol{u})\|p(\boldsymbol{u})] + \frac{1}{2}\sum_n \langle 1 + \log(m(x_n)) - m(x_n)\rangle_{q(x_n)}$$

$$+ \sum_n \int_{\boldsymbol{u},x_n,f(x_n)} q(x_n)q(\boldsymbol{u})q(f(x_n)|x_n,\boldsymbol{u})\log p(y_n|f(x_n)).$$

We can obtain the collapsed bound by noting that the optimal form for $q(\boldsymbol{u})$ is given by

$$q(\boldsymbol{u}) \propto p(\boldsymbol{u})\exp\left(\langle\log\mathcal{N}(\boldsymbol{y}; \mathbf{K_{fu}}\mathbf{K_{uu}^{-1}}\boldsymbol{u}, \sigma^2\mathbf{I})\rangle_{q(\boldsymbol{x})}\right).$$

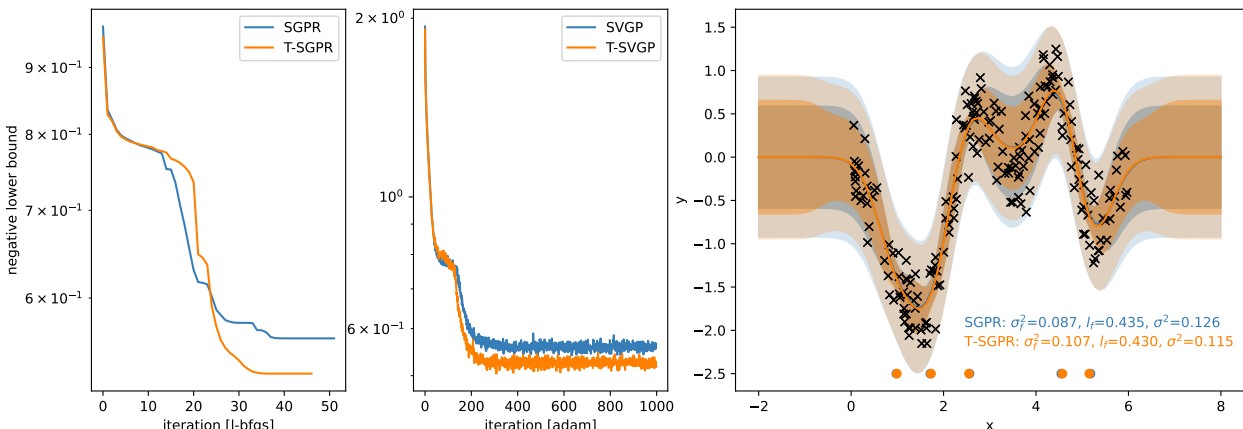

Figure 1: Left and middle: Optimisation traces for SGPR, T-SGPR, SVGP and T-SVGP on the Snelson dataset with 5 inducing points. Right: Predictive means and uncertainties. The stronger shade is for noiseless predictions.

Note also that

$$\int_{\boldsymbol{u},\boldsymbol{f}} q(\boldsymbol{u})q(\boldsymbol{f}|\boldsymbol{u})\log p(\boldsymbol{y}|\boldsymbol{f}) = \int_{\boldsymbol{u}} q(\boldsymbol{u})\log \mathcal{N}(\boldsymbol{y}; \mathbf{K}_{\mathbf{fu}}\mathbf{K}_{\mathbf{uu}}^{-1}\boldsymbol{u}, \sigma^2\mathbf{I}) - \sum_n \frac{m_n(x_n)d_n}{2\sigma^2}.$$

Together with Jensen's inequality, we arrive at the collapsed bound

$$\mathcal{F}(q(\boldsymbol{x})) = -\mathrm{KL}[q(\boldsymbol{x})\|p(\boldsymbol{x})] - \frac{1}{2}\sum_n \left\langle \frac{m_n(x_n)d_n}{2\sigma^2} - 1 - \log m_n(x_n) + m_n(x_n) \right\rangle_{q(x_n)}$$
$$+ \log\left(\int_{\boldsymbol{u}} e^{\langle \log \mathcal{N}(\boldsymbol{y};\mathbf{K}_{\mathbf{fu}}\mathbf{K}_{\mathbf{uu}}^{-1}\boldsymbol{u},\sigma^2\mathbf{I})\rangle_{q(\boldsymbol{x})}} p(\boldsymbol{u})\right).$$

Setting derivatives w.r.t. $m_n(x)$ to 0 gives

$$\langle m_n(x_n)\rangle_{q(x_n)} = \left\langle \frac{\sigma^2}{d_n + \sigma^2} \right\rangle_{q(x_n)}$$

which is satisfied by $m_n(x_n) = \frac{\sigma^2}{d_n+\sigma^2}$ or $m_n(x_n) = \left\langle \frac{\sigma^2}{d_n+\sigma^2}\right\rangle_{q(x_n)}$. The former is easier to implement as we do not need to (approximately) integrate out $x_n$ to find $m_n$.

## 6 Experimental results

We validate the utility of the proposed variational posterior in a suite of experimental settings. We switch the variational objective with the proposed approximation in each setting, keep all other configurations unchanged, and measure the two's predictive performance. Implementations based on GPytorch and GPflow are released in this repository `https://github.com/thangbui/tighter_sparse_gp`.

### 6.1 Toy 1-D regression

To build intuition about the proposed method's behaviour, we first evaluate it on a 1-D regression problem used by Snelson & Ghahramani (2005). We compare (i) Titsias' collapsed bound in eq. (5) [SGPR] with the proposed collapsed bound in eq. (9) [T-SGPR], and (ii) Titsias' uncollapsed bound [SVGP] with the proposed uncollapsed bound in eq. (8) [T-SVGP]. Figure 1 illustrates the optimisation trajectories of these methods and the final fits for both SGPR and T-SGPR using five inducing points. The final values for both

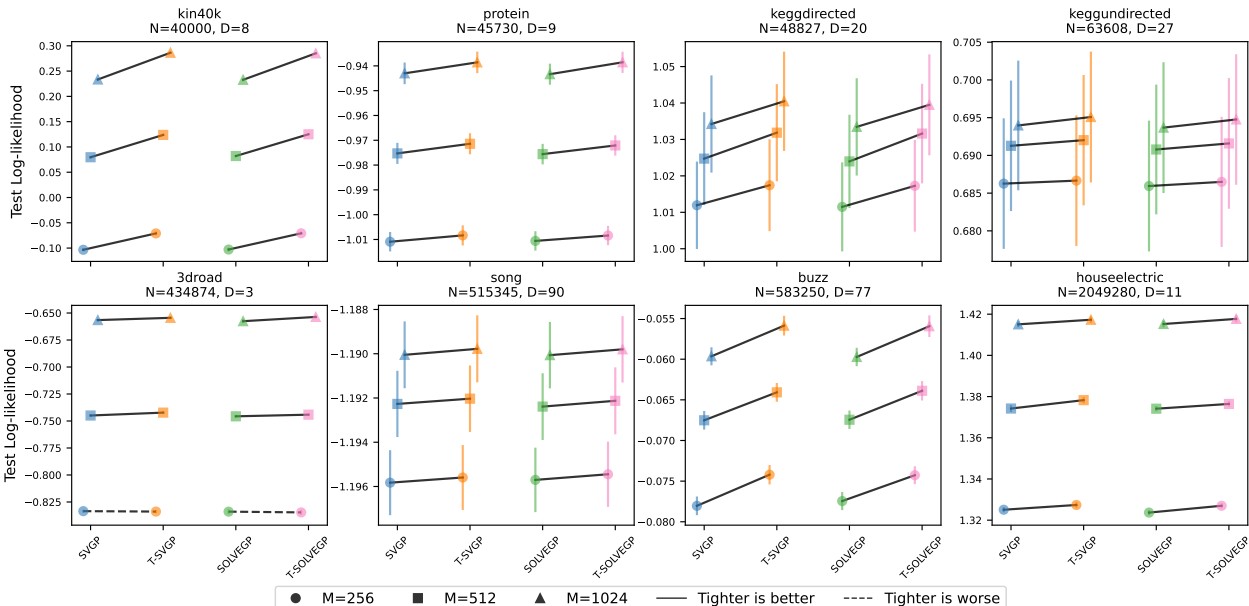

Figure 2: Test log-likelihood for various sparse approximations on eight regression datasets and various numbers of pseudo-points. For SOLVEGP and T-SOLVEGP, M is evenly split for $\boldsymbol{u}$ and $\boldsymbol{v}$. Higher is better. Best viewed in colour.

uncollapsed and collapsed versions of the proposed bound appear tighter than that of the Titsias' bound in practice. The learned hyperparameters reveal that T-SGPR prefers smaller observation noise (0.115) and larger kernel variance (0.107) compared to that of SGPR (0.126 and 0.087, respectively).

## 6.2 Efficient predictive variances

A key practical consideration is the computational cost of predictive variance in eq. (12). The exact computation requires $\mathbf{D}_{\mathbf{ff}}^{-1}$ which scales poorly with the training set size. We evaluate a simplified variant that omits the term that involves $\mathbf{D}_{\mathbf{ff}}$, the last term in eq. (12), and compare it to the exact variance calculation across three small benchmark datasets: wine, solar, and pumadyn32nm. Table 1 presents a detailed comparison between the exact and approximate versions. We can see a pattern across all datasets: the simplified variant consistently matches the full model's performance while offering substantial computational savings. For this reason, we will be using the simplified version for all remaining experiments. The improvements in predictive performance in sections 6.3 and 6.4 are therefore solely due to better estimation of the hyperparameters and a different $q(\boldsymbol{u})$.

| Dataset | N/D | eq. (12) | RMSE | Log-likelihood | Time (s) |
|---|---|---|---|---|---|
| wine | 1599/11 | w. last term | $0.47 \pm 0.01$ | $-0.66 \pm 0.01$ | $0.15 \pm 0.00$ |
| | | wo. last term | $0.47 \pm 0.01$ | $-0.66 \pm 0.01$ | $0.03 \pm 0.00$ |
| solar | 1066/10 | w. last term | $0.93 \pm 0.07$ | $-1.57 \pm 0.20$ | $0.07 \pm 0.00$ |
| | | wo. last term | $0.93 \pm 0.07$ | $-1.56 \pm 0.20$ | $0.03 \pm 0.00$ |
| pumadyn32nm | 8192/32 | w. last term | $1.00 \pm 0.01$ | $-1.42 \pm 0.01$ | $21.12 \pm 0.06$ |
| | | wo. last term | $1.00 \pm 0.01$ | $-1.42 \pm 0.01$ | $0.05 \pm 0.00$ |

Table 1: RMSE, log-likelihood, and run time for two variants of predictive variance computation.

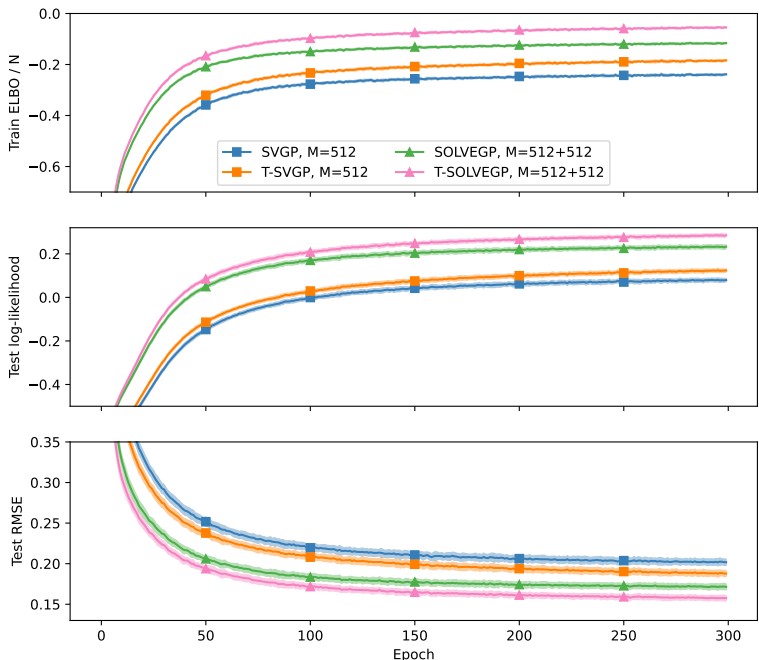

Figure 3: Variational bound and test performance for various approximations trained on the kin40k dataset. Best viewed in colour.

### 6.3 Large-scale regression benchmarks

We next compare four methods, SVGP, T-SVGP, SOLVEGP, and the SOLVEGP variant in eq. (14) [T-SOLVEGP], across three inducing-point configurations ($M = 256, 512, 1024$), on eight medium to large regression datasets. The datasets range from 40K to 2M data points with varying input dimensionalities (Yang et al., 2015). We use the Matern-3/2 kernel and repeat each experiment 10 times, each employing a random train/test split. The comparison results are shown in figs. 2, 7 and 8. We note that (i) both T-SVGP and T-SOLVEGP consistently match or slightly outperform (on 5/8 datasets), or significantly outperform (on 3/8 datasets) their base counterparts, (ii) the performance improvement is also consistent across various inducing-point configurations, (iii) the improvements (on 3/8 datasets) are also consistent across training runs and iterations, as shown in fig. 3 for the kin40k dataset, and (iv) SVGP and T-SVGP (and similarly SOLVEGP and T-SOLVEGP) have almost identical run time so the improvements here do not come at any cost. We also quantitatively compare the hyperparameters provided by these approximations, including the kernel variance, kernel lengthscale, and observation noise in figs. 9 to 11, respectively. Similar to the 1D example above, we observe that the proposed approximations prefer smaller observation noises and larger kernel variances.

### 6.4 MNIST classification

To evaluate the performance of the proposed approximation on non-Gaussian likelihoods, we run an experiment on the MNIST digit classification task with 256, 512, 1024, and 2048 inducing points, using the SVGP, T-SVGP, SOLVEGP, and T-SOLVEGP variational objectives. Figure 4 shows that both the proposed approximations achieve substantial performance gains in all metrics compared to their base versions.

### 6.5 Comparison to exact GP regression

We further evaluate the proposed approximation by comparing it to exact GP regression on a small dataset, following the set-up in (Titsias, 2009). We use the Boston housing dataset,[2] vary the number of inducing

---

[2]https://www.cs.toronto.edu/~delve/data/boston/bostonDetail.html

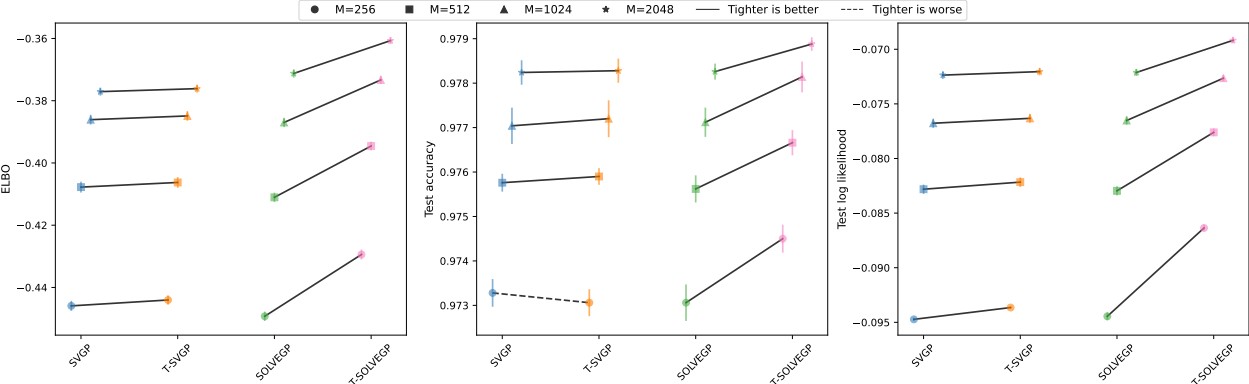

Figure 4: Log marginal likelihood approximations and test performance on the MNIST 10-way classification task. Best viewed in colour.

points, and use the collapsed bounds in eq. (6) [SGPR] and eq. (9) [T-SGPR]. The hyperparameters are fixed to that of the exact GP regression model so that the only variational parameter is the inducing inputs. We use four metrics for this comparison: the variational bound, test error, test log-likelihood, and the KL divergence between the predictive distributions given by exact GP regression and that provided by SGPR or T-SGPR. The average results across five repeats are included in fig. 5, confirming that the proposed tighter approximation consistently gives predictive distributions closer to the exact GPR predictions than that of SGPR. The test metrics that only use the marginal predictive distributions for T-SGPR are also better, consistent with the trend that, for fixed hyperparameters, a tighter variational approximation is better.

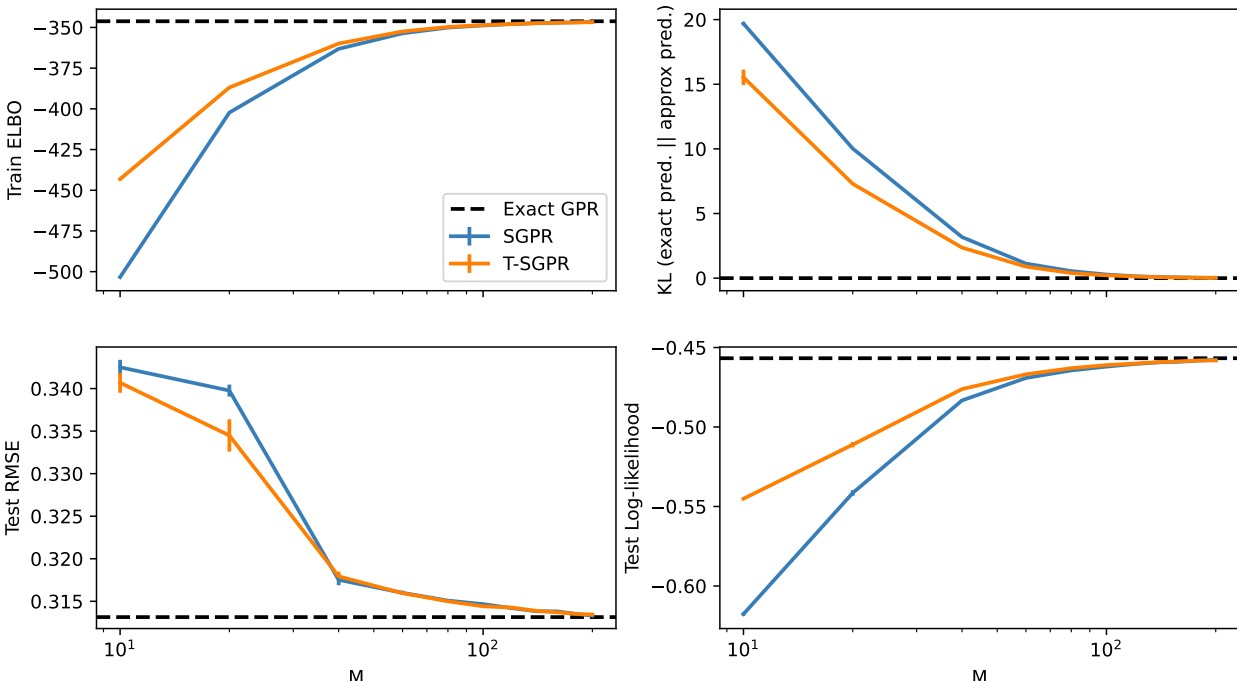

Figure 5: Comparing the Titsias' collapsed bound and the proposed collapsed bound to exact GP regression, with the model hyperparameters fixed to those of exact GP. Best viewed in colour.

In addition, we evaluate the impact of the hyperparameters on the bound and predictive performance when the variational approximation is fixed. Specifically, we optimise the uncollapsed SVGP bound on the Boston

housing dataset until convergence then subsequently switch to the tighter bound while keeping $q(\boldsymbol{u})$ and the inducing inputs fixed. The results included in fig. 6 demonstrate that the hyperparameters given by the improved bound yield marginally better predictive performance.

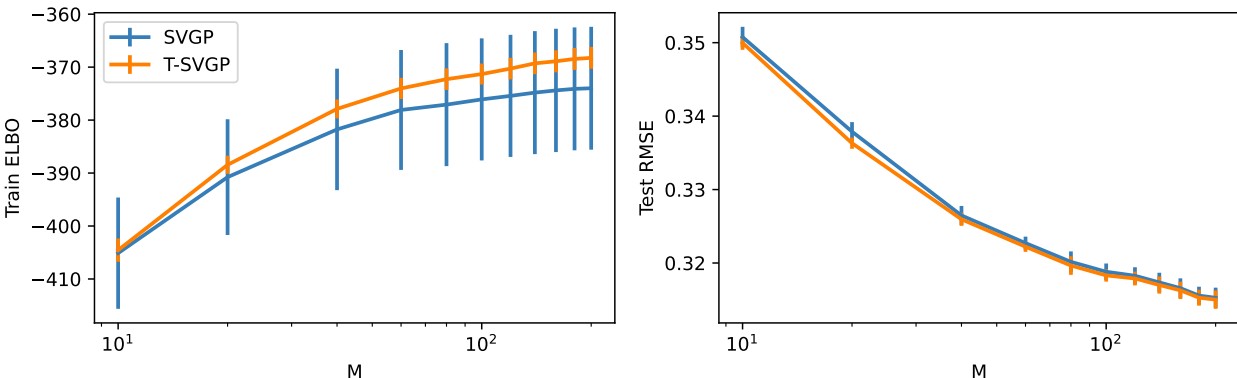

Figure 6: Comparing the SVGP bound and the proposed bound while keeping the variational distribution fixed. Best viewed in colour.

## 7 Summary

We build upon the standard sparse variational Gaussian process (SVGP) approximate posterior distribution through a simple modification to the conditional GP prior distribution at observed inputs. Using our proposed posterior approximation, we derive a collapsed bound which improves upon existing SVGP lower bounds to the log marginal likelihood, and an uncollapsed form which facilitates its application with non-Gaussian likelihoods and is compatible with stochastic mini-batch optimisation. Furthermore, we show how our approach can be used to improve non-standard SVGP posterior approximations, such as SOLVE-GP (Shi et al., 2020).

Our empirical results demonstrate consistent improvements in both predictive performance and log marginal likelihood estimates across diverse applications, including regression, classification, and latent variables modelling tasks. The proposed posterior approximations can be easily applied to other settings such as deep GPs and convolutional GPs (Van der Wilk et al., 2017; Blomqvist et al., 2020; Sun et al., 2021; Bui et al., 2016; Salimbeni & Deisenroth, 2017).

### Acknowledgments

We thank Martin Jankowiak for pointing out a parameterisation issue in an earlier version of this paper. We also thank the TMLR editor and reviewers for the constructive feedback.

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

## A  An even tighter but expensive approximation

We consider a more general form for the conditional covariance of $q(\boldsymbol{f}|\boldsymbol{u})$ as follows:

$$q(f) = p(f_{\neq \boldsymbol{f}, \boldsymbol{u}}|\boldsymbol{f}, \boldsymbol{u})q(\boldsymbol{f}|\boldsymbol{u})q(\boldsymbol{u}),$$
$$q(\boldsymbol{f}|\boldsymbol{u}) = \mathcal{N}(\boldsymbol{f}; \mathbf{K_{fu}}\mathbf{K_{uu}^{-1}}\boldsymbol{u}, \mathbf{C}),$$

Again, we can also obtain the optimal form for $q(\boldsymbol{u}) \propto p(\boldsymbol{u})\mathcal{N}(\boldsymbol{y}; \mathbf{K_{fu}}\mathbf{K_{uu}^{-1}}\boldsymbol{u}, \sigma^2 \mathbf{I})$, leading to the following collapsed bound

$$\mathcal{F}_6(\theta) = -\frac{N}{2}\log(2\pi) - \frac{1}{2}\boldsymbol{y}^\mathsf{T}(\mathbf{Q_{ff}} + \sigma^2\mathbf{I})^{-1}\boldsymbol{y} - \frac{1}{2}\log|\mathbf{Q_{ff}} + \sigma^2\mathbf{I}| - \frac{1}{2}\mathrm{trace}[(\sigma^{-2}\mathbf{I} + \mathbf{D_{ff}^{-1}})\mathbf{C}] - \frac{1}{2}\log|\mathbf{C}^{-1}\mathbf{D_{ff}}|.$$

We can derive the optimal $\mathbf{C}$, $\mathbf{C}^{-1} = \mathbf{D_{ff}^{-1}} + \sigma^{-2}\mathbf{I}$ and the bound becomes:

$$\begin{aligned}
\mathcal{F}_8(\theta) &= -\frac{N}{2}\log(2\pi) - \frac{1}{2}\boldsymbol{y}^\mathsf{T}(\mathbf{Q_{ff}} + \sigma^2\mathbf{I})^{-1}\boldsymbol{y} - \frac{1}{2}\log|\mathbf{Q_{ff}} + \sigma^2\mathbf{I}| - \frac{1}{2}\log|\mathbf{I} + \sigma^{-2}\mathbf{D_{ff}}| \\
&= -\frac{N}{2}\log(2\pi) - \frac{1}{2}\boldsymbol{y}^\mathsf{T}(\mathbf{Q_{ff}} + \sigma^2\mathbf{I})^{-1}\boldsymbol{y} - \frac{1}{2}\log|\mathbf{Q_{ff}} + \sigma^2\mathbf{I}| - \frac{1}{2}\sum_n \log(1 + \sigma^{-2}\lambda_n(\mathbf{D_{ff}})|,
\end{aligned}$$

where $\lambda_n(\mathbf{X})$ is the $n$-th eigenvalue of $\mathbf{X}$. The bound above is as expensive as the original log marginal likelihood.

# B Exploring alternative parameterisations for the conditional posterior

Instead of the general $\mathbf{C}$ as above or the form considered in the main text $\mathbf{C} = \mathbf{D_{ff}^{1/2}}\mathbf{M}\mathbf{D_{ff}^{\mathsf{T}/2}}$, we consider two other parameterisations that might allow efficient collapsed/un-collapsed bounds and predictions. We first rewrite the uncollapsed bound and the predictive mean and variance here for clarity,

$$\mathcal{F}_{\text{uncollapsed}} = -\mathrm{KL}[q(\boldsymbol{u})||p(\boldsymbol{u})] - \int_{\boldsymbol{u}} q(\boldsymbol{u})\mathrm{KL}[q(\boldsymbol{f}|\boldsymbol{u})||p(\boldsymbol{f}|\boldsymbol{u})] + \sum_n \int_{\boldsymbol{u}, f(x_n)} q(\boldsymbol{u})q(f(x_n)|\boldsymbol{u})\log p(y_n|f(x_n)),$$

$$m_* = \mathbf{k_{*u}}\mathbf{K_{uu}^{-1}}\mathbf{m}_{\boldsymbol{u}},$$

$$v_* = \mathbf{k_{**}} - \mathbf{k_{*u}}\mathbf{K_{uu}^{-1}}\mathbf{k_{u*}} + \mathbf{k_{*u}}\mathbf{K_{uu}^{-1}}\mathbf{S}_{\boldsymbol{u}}\mathbf{K_{uu}^{-1}}\mathbf{k_{u*}} - (\mathbf{k_{*f}} - \mathbf{Q_{*f}})(\mathbf{D_{ff}} - \mathbf{C})(\mathbf{k_{f*}} - \mathbf{Q_{f*}}),$$

We first consider $\mathbf{C} = \mathbf{M}^{1/2}\mathbf{D_{ff}}\mathbf{M}^{1/2}$. While this allows efficient exact predictive marginal distributions at training points, the middle term in the bound is costly to compute due to the presence of $\mathbf{D_{ff}}$:

$$-\int_{\boldsymbol{u}} q(\boldsymbol{u})\mathrm{KL}[q(\boldsymbol{f}|\boldsymbol{u})||p(\boldsymbol{f}|\boldsymbol{u})] = -\frac{1}{2}\mathrm{trace}(\mathbf{D_{ff}^{-1}}\mathbf{M}^{1/2}\mathbf{D_{ff}}\mathbf{M}^{1/2}) + \frac{1}{2}\log|\mathbf{M}| + \frac{N}{2}.$$

Another special case of the parameterisation presented in the main text is $\mathbf{C} = m\mathbf{D_{ff}}$, i.e., a single $m$ is shared across all training points. This conveniently leads to tractable exact predictive variances at training points, $v_n = md_n + \mathbf{k}_{f_n \boldsymbol{u}}\mathbf{K_{uu}^{-1}}\mathbf{S}_{\boldsymbol{u}}\mathbf{K_{uu}^{-1}}\mathbf{k}_{\boldsymbol{u}f_n}$. The middle term in the bound can be simplified to,

$$-\int_{\boldsymbol{u}} q(\boldsymbol{u})\mathrm{KL}[q(\boldsymbol{f}|\boldsymbol{u})||p(\boldsymbol{f}|\boldsymbol{u})] = \frac{N}{2}[1 + \log(m) - m].$$

In the regression case, this leads to the optimal $m = \sigma^2/(N^{-1}\sum_n d_n + \sigma^2)$ and the following collapsed bound,

$$\mathcal{F}_5(\theta) = -\frac{N}{2}\log(2\pi) - \frac{1}{2}\boldsymbol{y}^\mathsf{T}(\mathbf{Q_{ff}} + \sigma^2\mathbf{I})^{-1}\boldsymbol{y} - \frac{1}{2}\log|\mathbf{Q_{ff}} + \sigma^2\mathbf{I}| - \frac{N}{2}\log\left(1 + \frac{\sum_n d_n}{N\sigma^2}\right). \quad (15)$$

This is identical to the bound derived by Artemev et al. (2021). As shown in the main text, this bound is looser than the collapsed bound in eq. (9), due to the Jensen's inequality $\log(1+\sum_n x_n/N) \geq N^{-1}\sum_n \log(1 + x_n)$.

# C Additional experimental results

## C.1 Large-scale regression benchmarks

In addition to the test log likelihood results presented in the main text, we also compare the approximations using the test root mean squared error (RMSE) in fig. 7 and the variational lower bound in fig. 8. The

proposed approximations (T-SVGP and T-SOLVEGP) tend to give smaller RMSEs and better log marginal likelihood approximations, indicating tighter bounds leading to better predictions.

We also quantitatively compare the hyperparameters provided by these approximations, including the kernel variance, kernel lengthscale, and observation noise in figs. 9 to 11, respectively. Overall, the proposed approximations prefer smaller observation noises and larger kernel variances.

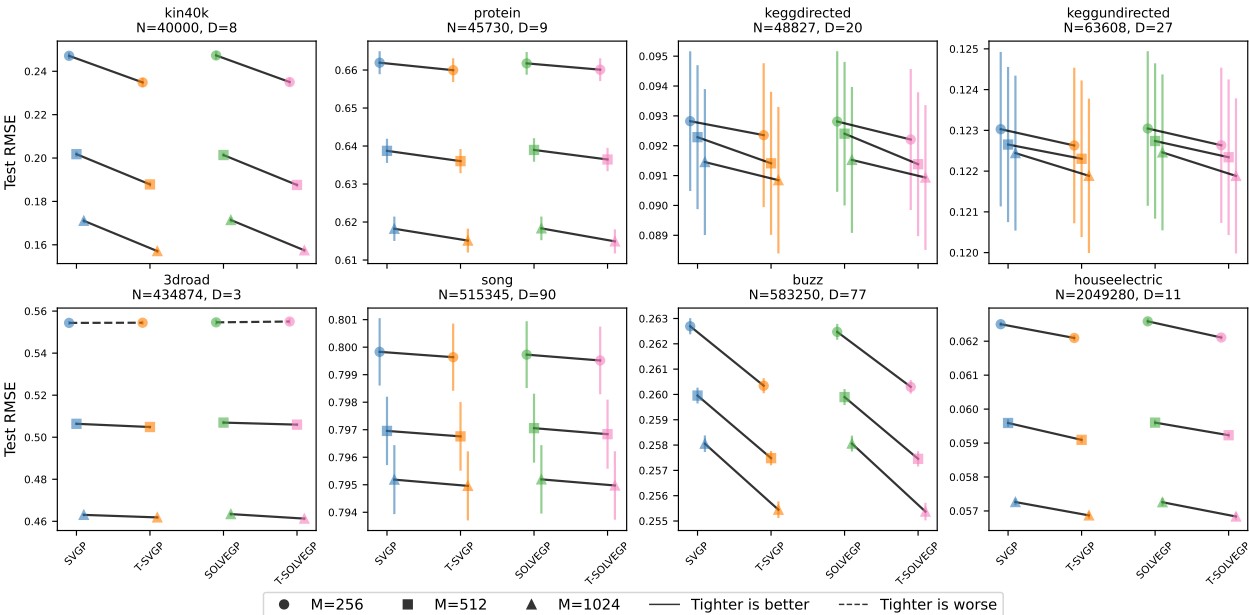

Figure 7: Test root mean squared errors (RMSE) for various sparse approximations on eight regression datasets and various numbers of pseudo-points. For SOLVEGP and T-SOLVEGP, M is evenly split for $\boldsymbol{u}$ and $\boldsymbol{v}$. Lower is better. Best viewed in colour.

## C.2 $m_n$ parameterisations for non-Gaussian likelihoods

We next assess the difference between two parameterisations of $m_n$, $m_n = \beta/(d_n + \beta)$ with $\beta$ shared across all data points, and $\{m_n\}_{n=1}^N$ with each $m_n$ specific to a training point, on a simple binary classification dataset. The results are included in fig. 12. We observe that (i) the former parameterisation is tighter than vanilla SVGP and the latter parameterisation is even tighter, and (ii) the difference between the predictive probabilities given by these parameterisations is small. For small and medium datasets, it is thus potentially beneficial to have $N$ additional parameters. For large datasets, the former parameterisation is preferred to keep the number of parameters manageable.

## C.3 GPLVM on the oil flow dataset

Finally, we demonstrate the proposed method's applicability to latent variable models through experiments with Bayesian GPLVM on the oil flow dataset. The multi-phase oil flow dataset consists of 1000, 12-dimensional data points belonging to three classes which correspond to the different phases of oil flow in a pipeline (Bishop & James, 1993). Figure 13 compares the standard variational BGPLVM (Damianou et al., 2016; Lalchand et al., 2022) [V-BGPLVM] against the proposed approximation in section 5 [TV-BGPLVM], averaged across five repeats. The optimisation trajectories show that TV-BGPLVM achieves a lower final negative ELBO (roughly $-5.5$ versus $-5.2$), indicating a more accurate posterior approximation.

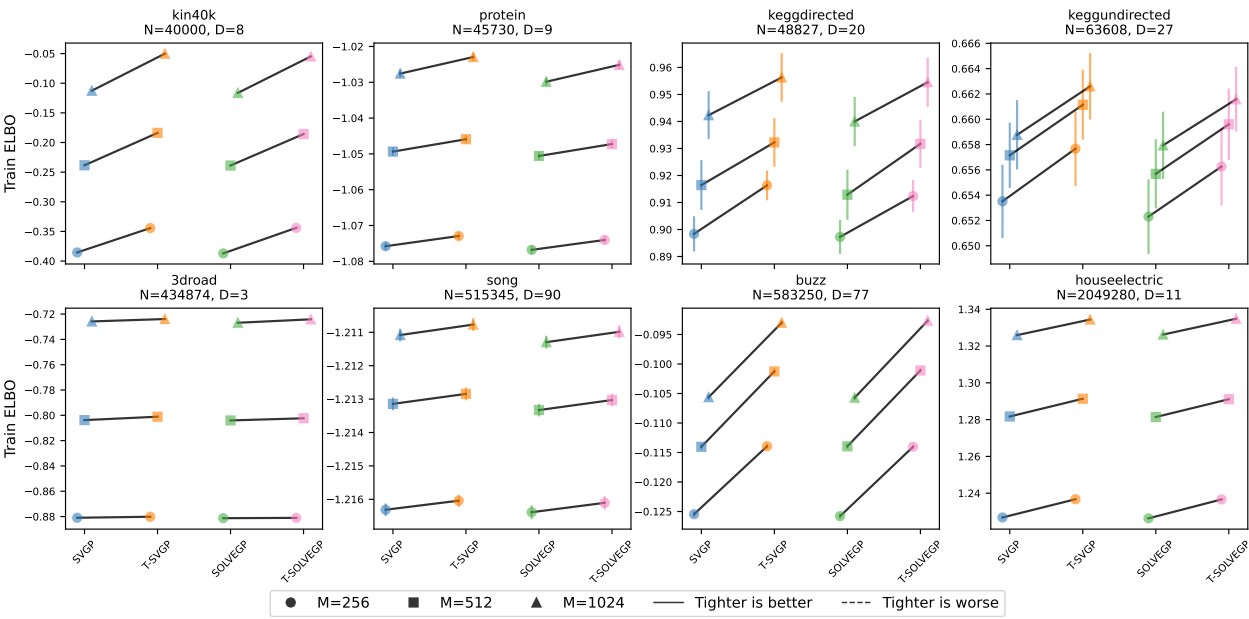

Figure 8: Log marginal likelihood approximations (ELBO) for various sparse approximations on eight regression datasets and various numbers of pseudo-points. For SOLVEGP and T-SOLVEGP, M is evenly split for $\boldsymbol{u}$ and $\boldsymbol{v}$. Higher is better. Best viewed in colour.

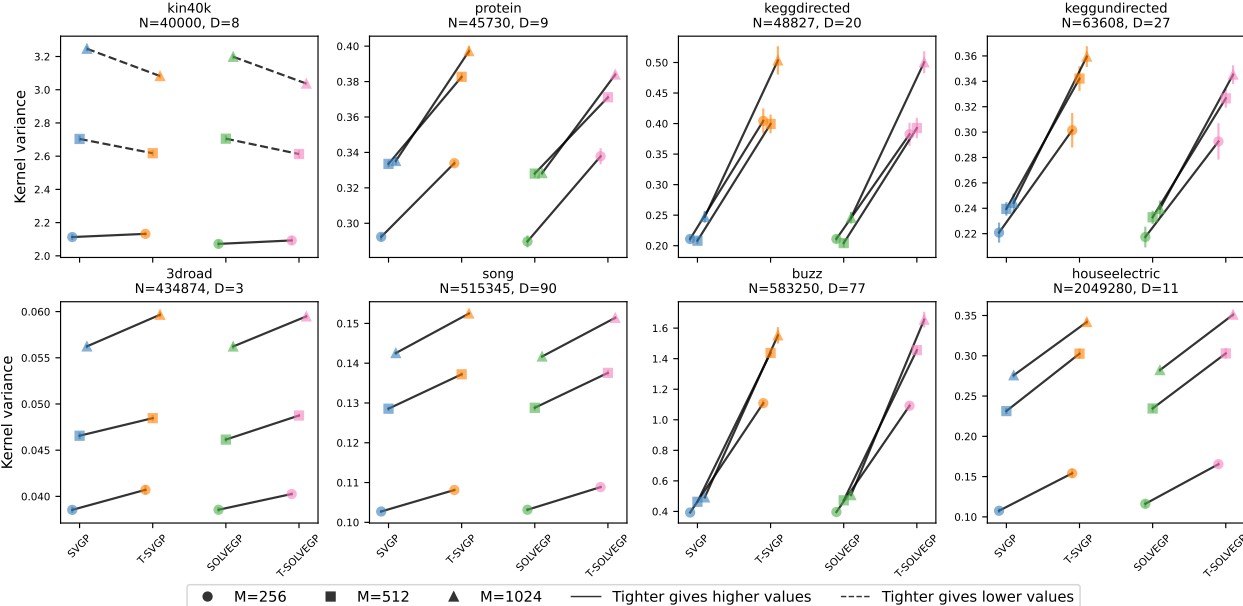

Figure 9: Kernel variances for various sparse approximations on eight regression datasets and various numbers of pseudo-points. For SOLVEGP and T-SOLVEGP, M is evenly split for $\boldsymbol{u}$ and $\boldsymbol{v}$. Best viewed in colour.

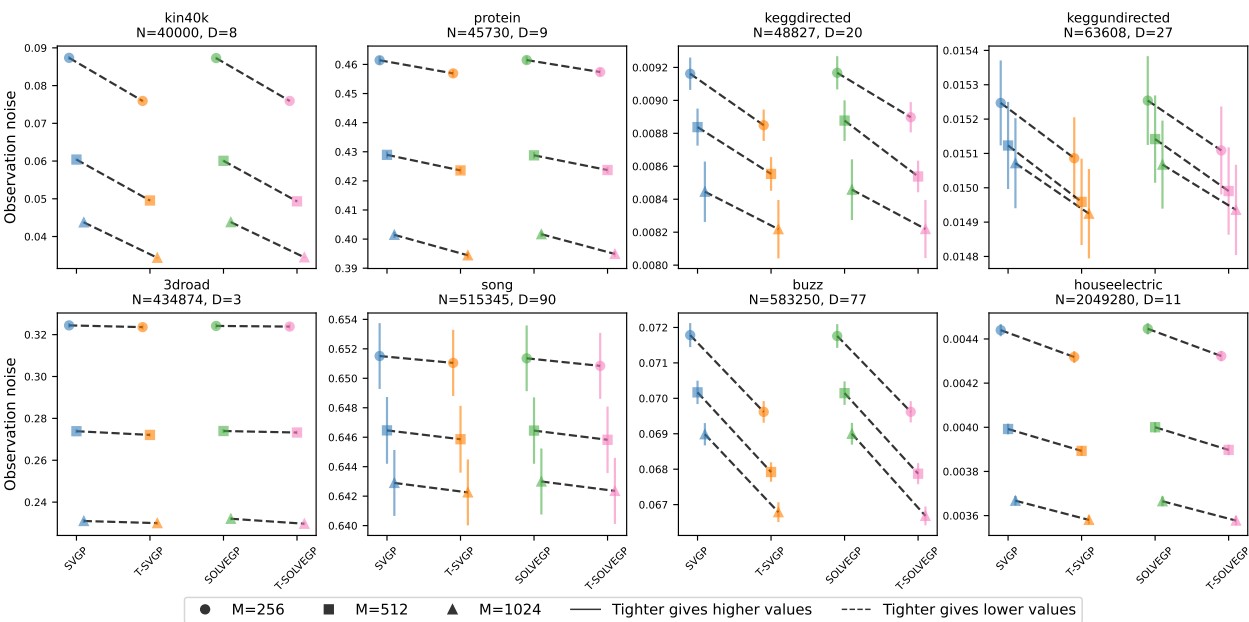

Figure 10: Observation noise variances for various sparse approximations on eight regression datasets and various numbers of pseudo-points. For SOLVEGP and T-SOLVEGP, M is evenly split for $\boldsymbol{u}$ and $\boldsymbol{v}$. Best viewed in colour.

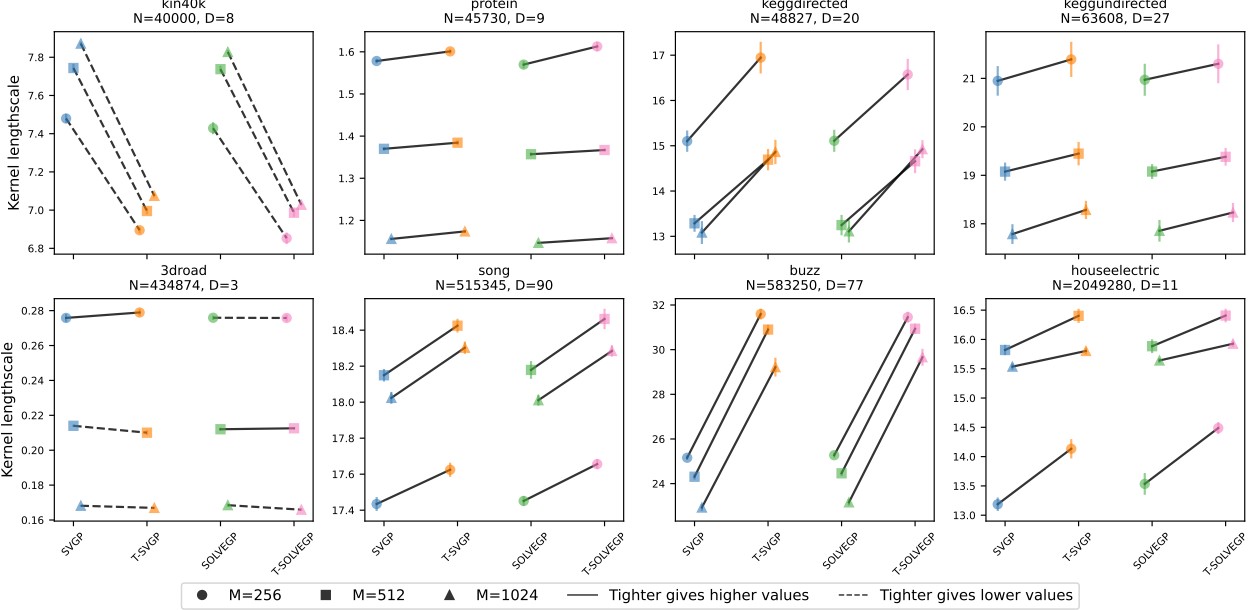

Figure 11: Kernel lengthscales for various sparse approximations on eight regression datasets and various numbers of pseudo-points. For SOLVEGP and T-SOLVEGP, M is evenly split for $\boldsymbol{u}$ and $\boldsymbol{v}$. Best viewed in colour.

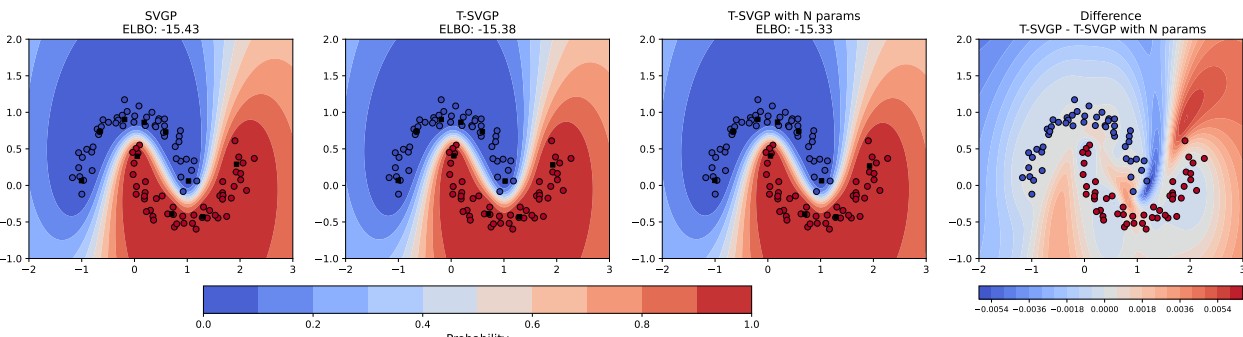

Figure 12: A comparison of $m_n$ parameterisations on a two-dimensional binary classification task. First: predictions when using the standard SVGP approximation. Second: predictions when using the proposed tighter SVGP bound with $m_n = \beta/(d_n + \beta)$ with $\beta$ shared across all data points. Third: predictions when using the proposed tighter bounds with $N$ additional parameters $\{m_n\}_{n=1}^N$. Fourth: the difference between the predictive probabilities given by the two $m_n$ parameterisations.

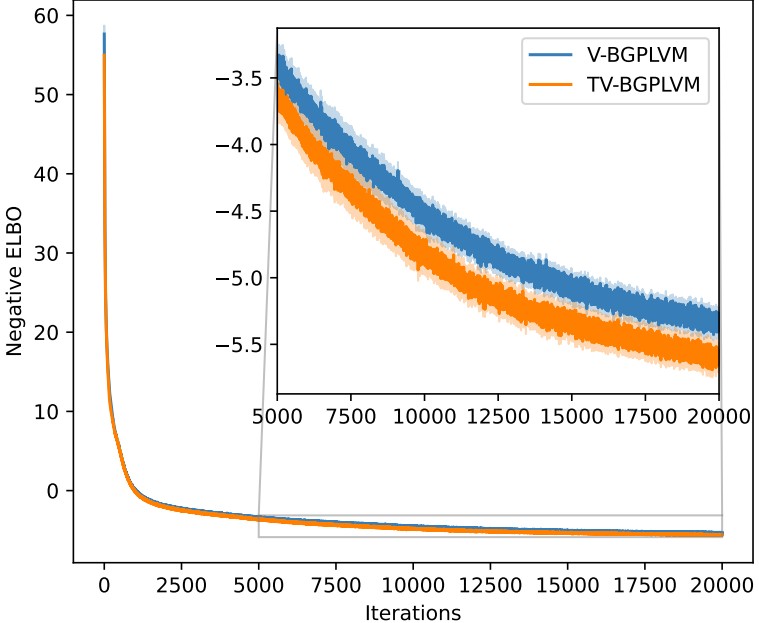

Figure 13: Optimisation traces for variational Bayesian GPLVM on the oil flow dataset. Best viewed in colour.

