# OpenReview forum: "Tighter sparse variational Gaussian processes"
_TMLR — Accepted by TMLR_

### Review · Reviewer_uPL1 · 2025-02-23

**Summary Of Contributions:**

This paper presents a sparse GP learning method based on the inducing point framework and formulates a tighter variational bound. This tighter bound is achieved by allowing the variational posterior at the training inputs q(f|u) to have weaker dependency on the inducing points q(u). Similar to standard inducing point methods, the paper a collapsed bound and a non-collapsed bound, where the former applies to Gaussian likelihoods and the latter is amenable to non-Gaussian likelihoods and mini-batch optimization. The paper also discusses the application of the proposed method to handle orthogonally structured inducing points (Shi et al) and GP latent variable models.

Experiments on regression, classification, and latent variable models show that the proposed method were able to achieve a better training ELBO compared to standard inducing point methods, and match or outperform their test-time predictive performance.

**Audience:**

Yes

**Claims And Evidence:**

Yes

**Requested Changes:**

A discussion over the weakness point raised above would be appreciated.

**Strengths And Weaknesses:**

Strengths:
- The paper is well-written and easy-to-follow.
- The proposed tighter bound is a novel contribution, and provides a new, orthogonal perspective to improve upon existing variational GP methods.
- The experiment validation is comprehensive and convincing, which shows consistent improvement by leveraging a tighter variational bound.

Weakness:
-  While the paper focuses on the predictive performance, there is a lack of discussion on the implication of such tighter bounds in other aspects, e.g. uncertainty quantification. For example, Bauer et al 2016 shows that various sparse GP methods can overestimate/underestimate the noise parameter. It remains unclear whether the introduced term $m_n = \frac{\sigma^2}{d_n+\sigma^2}$, which appears to function as a shrinking operator, offers any interpolation effects in uncertainty quantification between the SVGP and some limiting class of GP approximation models. Expanding on this aspect could offer valuable insights for using GP in downstream applications beyond predictive performance.

Reference:
- Bauer et al Understanding Probabilistic Sparse Gaussian Process Approximations. Neurips 2016.

---

> ### Author Response · Authors · 2025-03-10
> **Response**
>
> We thank the reviewer for the positive comments.
>
> We have now included several plots in the appendix showing the hyperparameters obtained by optimising the new variational bound. Specifically, the observation noise tends to be smaller, and the kernel variance tends to be larger than those of vanilla sparse GPs.
>
> We have also run an experiment on a small dataset to understand the benefits of the new bounds when the variational parameters or the hyperparameters are fixed. The results are now in section 6.5 in the main text.
>
> Please let us know if you have further thoughts.

---

### Review · Reviewer_Zy61 · 2025-03-02

**Summary Of Contributions:**

Starting from the variational framework for sparse Gaussian processes, the authors propose a different parametrization of the variational distribution $q(\\boldsymbol{f}|\\boldsymbol{u})$ that leads to a tighter lower bound on the model evidence. They replace the standard variational distribution $q(\\boldsymbol{f}|\\boldsymbol{u}) = p(\\boldsymbol{f}|\\boldsymbol{u}) = \\mathcal{N}(\\boldsymbol{K}\_{fu}\\boldsymbol{K}\_u^{-1}\\boldsymbol{u}, \\boldsymbol{K}\_f-\\boldsymbol{Q}\_f)$, where $\\boldsymbol{Q}\_f = \\boldsymbol{K}\_{fu}\\boldsymbol{K}\_u^{-1}\\boldsymbol{K}\_{fu}^\\intercal$, with a more general one $q(\\boldsymbol{f}|\\boldsymbol{u}) = \\mathcal{N}(\\boldsymbol{K}\_{fu}\\boldsymbol{K}\_u^{-1}\\boldsymbol{u}, \\boldsymbol{C})$ with different parametrizations of $\\boldsymbol{C}$. The authors highlight an approximation where $\\boldsymbol{C}$ is has the same singular value decomposition to $\\boldsymbol{K}\_f-\\boldsymbol{Q}\_f$ except that the singular values multiplied by positive free parameters $\\{m\_i\\}\_{i=1}^N\\subset\\mathbb{R}^N$.

However, the approximate posterior of the highlighted approximation still requires inverting matrices of rank $N$, which has been a roadblock for previous attempts at these reparametrizations. The authors propose effectively using the original prediction distribution as if $q(\\boldsymbol{f}|\\boldsymbol{u})=p(\\boldsymbol{f}|\\boldsymbol{u})$ while using the hyperparameter/variational parameter estimations from their method. Experiments on UCI datasets demonstrate this approach doesn't negatively impact predictive performance.

The authors extend their work beyond Gaussian process regression/classification to the Gaussian process latent variable model (GPLVM), a building block of the more expressive deep Gaussian process model, and to SOLVE-GP, another reparametrization of variational sparse GP.

These proposals are validated by proving that the ELBO of this new parametrization is tighter than standard approaches and by comparing predictive metrics and ELBO across methods. In most cases, the new method achieves either better or comparable ELBO/predictive metrics.

**Audience:**

Yes

**Broader Impact Concerns:**

I believe there are no obvious broader impact concerns with this paper.

**Claims And Evidence:**

Yes

**Requested Changes:**

- [Strengthen] It would be nice to have appendices A and B incorporated into section 3.
- [Critical] I believe section 6.5 is currently weak and should either be improved or moved to the appendix. For improvement: the text doesn't specify how many repetitions were performed for the oil experiment — if only one was conducted, the experiment should be repeated multiple times as in section 6.3; As mentioned earlier, additional metrics should be used, following Damianou et al. (2016), since a lower ELBO alone doesn't clearly communicate the downstream benefit of this methodology; Including another dataset or task for GPLVMs would also better illustrate the advantages of this approximation.

**Strengths And Weaknesses:**

Strengths:

- The proposals are well motivated and easy to follow. The authors also compare their approximation to two other bounds in the literature.
- A small ablation study was added to justify the cheaper predictive variances, thus allowing experimental validation of claims that cannot be theoretically validated.

Weaknesses:

- The section on experiments for GPLVM feels incomplete. Only the oil flow dataset is used, and the evaluation metric is limited to the ELBO value. This contrasts with other experiments that use predictive performance evaluation. Compare this with the exposition in Damianou et al. (2016), which uses nearest neighbor prediction as a proxy for class separation and additionally tests GPLVMs in various tasks such as image reconstruction.
- The authors could expand more on their proposal, which is currently split between the main text and appendices A and B. In particular, a discussion on $\boldsymbol{M}$ controlling the singular values of the conditional kernel, perhaps including how it might affect the spectral decomposition of the kernel, would be valuable.
- None of the metrics compare the quality of the approximation to the true distribution. Since the authors tested their method on small-scale datasets in section 6.2, it should be possible to replicate Titsias's (2009) analysis and compute the moment-matching KL between the approximate predictive distribution and the full GP predictive distribution, demonstrating that the tighter ELBO corresponds to a better approximation.

---

> ### Author Response · Authors · 2025-03-11
> **Response**
>
> We thank the reviewer for the constructive comments.
>
> On Section 6.5: we agree and have moved this to the appendix. We have added a comparison to exact GP regression as you suggested. Please see the latest revision.
>
> On appendices A and B: we hoped to have a self-contained description of the proposed approximation in the main text, but now agree with your recommendation to explicitly link these appendices to have a clearer big picture. We will rework this shortly.
>
> Please let us know if you have further comments.

---

> > ### Comment · Reviewer_Zy61 · 2025-04-08
> >
> > Dear authors,
> > Are there still plans on linking the appendices together or not?

---

> ### Author Response · Authors · 2025-04-08
>
> Yes, we will move Appendix A to the main text. We will also make it clear that $\mathbf{C} = \mathbf{D}\_{\mathbf{ff}}^{1/2} \mathbf{M} \mathbf{D}_{\mathbf{ff}}^{\top/2}$ makes things tractable, with $\mathbf{M} = \mathbf{I}$, $\mathbf{M} = m \mathbf{I}$, and $\mathbf{M} = diag([m_1, \dots, m_N])$ giving the original Titsias' bound, Artermev et al's bound, and our (and Titsias' concurrent work) bounds, respectively. Thanks for the follow-up! The lead author is on leave and will make these changes as soon as they return.

---

### Review · Reviewer_c3F7 · 2025-03-09

**Summary Of Contributions:**

The paper revisits the popular variational approximation for sparse variational Gaussian processes (GPs). It introduces an adjusted variational approximation that replaces the conditional posterior (traditionally set equal to the prior) with a more flexible variational distribution. The new approximation leads to a tighter variational bound. Furthermore, the approach naturally integrates into stochastic mini-batch optimization, enabling scalability to large datasets. Experiments validate improvements across multiple tasks, including regression, classification, and latent-variable modelling. The paper also demonstrates how the proposed method could benefit existing algorithms such as SOLVE-GP and GPLVM, highlighting its broad applicability.

**Audience:**

Yes

**Claims And Evidence:**

Yes

**Requested Changes:**

## Requested Critical changes

- The motivation for $q(\mathbf{f} \mid \mathbf{u})$ is abrupt. The start of section three goes straight into defining the new approximation. It would help the reader understand what goes wrong with the normal approximation. Some simple toy examples contrasting the predictive variances here would again help similarly as in [1]. I suppose this is the purpose of Figure 1, but it introduces a full optimization of hyperparameters and variational approximation, making it hard to tell the exact contribution. You could reuse this example but zoom in on the predictive variances, hold hyperparameters, and inducing points constant.

- Related to this, it's unclear how the hyperparameters interplay with the new approximation. The paper states 'solely due to better estimation of the hyperparameters and a different $q (\mathbf{u})$, but which one can you quantify this statement? What is the interplay with hyperparameters?

- Non-Gaussian likelihood approximation. The definition of $\beta$ needs some more explaining and testing. You are assuming that we can use the exact Gaussian approximation in the non-Gaussian case to save on blowing up the parameter count. I am okay with that, but it would be great to see a test. We know from [2] that the likelihood approximation now looks similar to a heteroskedastic regression, so a constant approximation would potentially adjust the term accordingly. However, the more expressive form could improve the non-Gaussian approximation. It's not clear, I don't think.

- I found the writing of the Predictions section confusing. Why can we ignore the last term? I think it comes back to the interplay of variational parameters and hyperparameters.

- A recent paper [3], appears to have come out concurrently. It would be good to mention and contrast the differences between the two.

- The $\mathbf{c}$ that appears before equation (9) is not defined. I understand it's a constant, but it would be good to say that.
'We not that $F_4(\theta)$' the F is not in the right latex format with eqn (9).


## General (non-critical) improvements

- Could you have $q(\mathbf{f} \mid \mathbf{u})$ be in a different color to contrast equation (7) with equation (4) to help the reader to distinguish the change quickly?

- The $\mathbf{c}$ that appears before equation (9) is not defined. I understand it's a constant, but it would be good to say that.
It would be interesting to plot the various terms in the new collapsed bound during optimization.

- The GPLVM on the oil flow example seems redundant as it only plots the ELBO. Can you not visualize the model's result in some way to see a change?


[1] Bauer et al, Understanding Probabilistic Sparse Gaussian Process Approximations (2016)

[2] Adam et al, Dual parameterization of sparse variational Gaussian processes (2021).

[3] Titsias, New Bounds for Sparse Variational Gaussian Processes  (2025).

**Strengths And Weaknesses:**

The main strength of the paper is that it attempts to improve the approximation of sparse variational Gaussian processes (SVGP), which is a key method in scaling up GPs. The extra GPLVM and SOLVE-GP applications solidify the technique's applicability. Furthermore, the experiments are convincing in benchmark datasets. The idea is well explained, and the paper is well written.

However, the motivation for the introduced conditional posterior $q(\mathbf{f} \mid \mathbf{u})$  could be clarified. The paper would benefit from a more explicit exploration of the interplay between hyperparameters and the proposed approximation, perhaps through controlled experiments isolating the effects. Moreover, explicitly demonstrating the constant-parameter approximation's validity in the non-Gaussian likelihood setting and discussing recent related works would strengthen the presentation.

---

> ### Author Response · Authors · 2025-03-11
> **Response**
>
> Thanks for your constructive feedback.
>
> On the interaction between the hyperparameters and variational parameters in the new bounds: Thanks for your suggestion. We have now added an experiment on a small dataset in section 6.5, comparing the approximations when the variational parameters or the hyperparameters are kept fixed. In both cases, we see an improvement in predictive performance. In addition, we also include the plots showing the hyperparameters on the regression datasets in the appendix. Consistent with the 1D example in Figure 1, the noise variance tends to be smaller, and the kernel variance tends to be larger when using the new bounds.
>
> On removing the last term of the predictive variances: this is an approximation, and unfortunately we don't have a strong theoretical argument for it. Empirically, removing this doesn't negatively impact the performance.
>
> On non-Gaussian likelihood approximation: Thanks for the suggestion on using per-datapoint parameters. We will check this and update the comment shortly. Update: we have included a small experiment in the appendix (see figure 12) to compare the two parameterisations.
>
> We have also fixed the typos. Please see the latest revision and let us know if you have further comments.

---

### Decision · Action_Editor_Fcnv · 2025-04-24

**Recommendation:** Accept as is

**Comment:**

This paper significantly improves one of the most popular scalable GP methods. Its experiments are thorough and broad, providing substantial evidence for this method. These comprehensive results should lead to the adoption of this method, provided that the authors follow through on pull requests to popular GP libraries.

The biggest concern with this paper was organization and presentation, as the original exposition of the method was not easy to follow. The authors worked with the reviewers to improve the flow and clarity of the paper, though they have not yet combined an appendix despite promising a reviewer that they would do so.

**Audience:**

This paper introduces a notable improvement to one of the most popular scalable GP approaches. The authors demonstrate its efficacy on various tasks (regression, classification, latent variable modelling) and discuss extending it to other related methods (orthogonally decomposed GPs, etc.). It will be a paper of interest to the GP community.

**Claims And Evidence:**

This paper offers a sound mathematical derivation of a tighter lower bound of the evidence of sparse Gaussian processes. While the presentation of the bound could be improved (as it is split between the main text and the append), it qualifies as clear evidence. The regression, classification, and latent variable model experiments demonstrate efficacy in numerous applications. The experiments show a tighter ELBO during training, validating the derivation of the method and increased predictive performance.